# In Vitro Probiotic Characterization and Safety Assessment of Lactic Acid Bacteria Isolated from Raw Milk of Japanese-Saanen Goat (*Capra hircus*)

**DOI:** 10.3390/ani13010007

**Published:** 2022-12-20

**Authors:** Yukimune Tanaka, Ni Putu Desy Aryantini, Eiki Yamasaki, Makoto Saito, Yui Tsukigase, Hirotaka Nakatsuka, Tadasu Urashima, Risa Horiuchi, Kenji Fukuda

**Affiliations:** 1Department of Life and Food Sciences, Obihiro University of Agriculture and Veterinary Medicine, 2-11 Inada-cho, Obihiro, Hokkaido 080-8555, Japan; 2Department of PR Science, PT Yakult Indonesia Persada, Jakarta Selatan, Jakarta 12530, Indonesia; 3Department of Veterinary Medicine, Obihiro University of Agriculture and Veterinary Medicine, 2-11 Inada-cho, Obihiro, Hokkaido 080-8555, Japan; 4Kisara Farm Cheese Factory, Haobiminami 10, Shimizu-cho, Kamikawa-gun, Hokkaido 089-0356, Japan; 5Research Center for Global Agromedicine, Obihiro University of Agriculture and Veterinary Medicine, 2-11 Inada-cho, Obihiro, Hokkaido 080-8555, Japan

**Keywords:** antipathogenic activity, antimicrobial resistance, cheese starter, *Lacticaseibacillus rhamnosus*

## Abstract

**Simple Summary:**

This is the first report of highly safe and potential probiotic strains, *Lacticaseibacillus rhamnosus* YM2-1 and YM2-3, isolated from the raw milk of dairy Japanese-Saanen goats. The antipathogenic activities of the two strains were identified based on their probiotic features. Although *L. rhamnosus* is on the list of qualified presumption of safety (QPS)-recommended biological agents assessed by the European Food Safety Authority (EFSA), additional in vitro examinations performed in this study ensure the safety of both strains, considering the risk of possible strain-dependent unfavorable characteristics.

**Abstract:**

Two novel probiotic strains of lactic acid bacteria were successfully isolated from the raw milk of dairy Japanese-Saanen goats. Selection criteria for positive candidates were grown on de Man–Rogosa–Sharpe or M17 selective medium at 30, 35, or 42 °C anaerobically, and characterized based on Gram reaction, catalase test, and tolerance to low pH and bile salts. Among the 101 isolated positive candidates, two strains, YM2-1 and YM2-3, were selected and identified as *Lacticaseibacillus rhamnosus* using 16S rDNA sequence similarity. Culture supernatants of the two strains exhibited antipathogenic activity against *Salmonella enterica* subsp. *enterica* serovar. Typhimurium, *Shigella sonnei*, methicillin-resistant *Staphylococcus aureus*, methicillin-sensitive *Staphylococcus aureus*, *Listeria monocytogenes*, and *Escherichia coli* O157. The antipathogenic activities were retained to some extent after neutralization, indicating the presence of antipathogenic substances other than organic acids in the culture supernatants. The two strains were sensitive with coincidental minimum inhibition concentrations (indicated in the parentheses hereafter) to ampicillin (0.25 μg/mL), chloramphenicol (4 μg/mL), gentamycin (4 μg/mL), kanamycin (64 μg/mL), streptomycin (16 μg/mL), and tetracycline (4 μg/mL). Furthermore, the two strains were resistant to clindamycin (16 μg/mL) and erythromycin (4 μg/mL). In addition, both YM2-1 and YM2-3 strains showed less unfavorable activities, including bile acid bioconversion, carcinogenic-related enzymes, mucin degradation, plasminogen activation, and hemolysis, than the detection limits of in vitro evaluation methods used in this study. In summary, *L. rhamnosus* YM2-1 and YM2-3 are highly safe and promising probiotic strains applicable in the dairy industry, and were first isolated from the raw milk of Japanese-Saanen goats.

## 1. Introduction

Lactic acid bacteria (LAB) are Gram-positive, rod- or sphere-shaped, non-spore-forming bacteria that secrete lactate by metabolizing more than 50% of the total intake of carbohydrates. They belong to the phylum Firmicutes, a diverse phylum consisting of more than 250 genera, including *Lactobacillus*. Recently, reclassification of the families *Lactobacillaceae* and *Leuconostocaceae* was performed on the basis of genome-wide information; hence, several species were reclassified accordingly [1]. LAB are ubiquitous in the natural ecological niche, and they are symbionts of plants [2,3], insects [4,5], and vertebrates [6,7]. Regarding human activities, LAB are used for food processing and preservation, namely, fermentation. Organic acids such as lactate, acetate, and butyrate secreted by LAB decrease the pH of the surrounding environment, resulting in the suppression of the growth of food-spoilage bacteria. For instance, milk fermentation is assumed to have occurred by the action of LAB coincidentally present in raw milk or its container, which is almost simultaneous with the time when cow domestication started, approximately between 7000 and 17,000 years ago [8,9]. In addition to the advantages of LAB in food preservation, some LAB exhibit beneficial health properties, such as antimicrobial [10,11], antidiabetic [12,13], hypotensive [14,15], and immunostimulatory activities [16,17,18], as well as diarrheal relief [19]. In addition to organic acids [20,21,22], bacteriocins [23,24,25], exopolysaccharides [26,27,28], and glycolipids [29] are beneficial metabolites obtained from LAB.

Fermented foods are undoubtedly the best source for exploring beneficial LAB. Indeed, several LAB strains have been isolated to date from fermented foods, such as pickles [30,31], beverages [32,33], yogurts [25,34,35,36], cheeses [25,37,38,39,40], and sausages [41]. In addition, the host intestines and feces [7,21,42], insects [4], and marine organisms [7] are of interest to researchers. Despite the relatively low population of LAB, raw milk from animals is also an attractive source of probiotics, which is defined as “live microorganisms which when administrated in adequate amounts confer a health benefit on the host” by FAO/WHO [43]. Milk, genital, and excretory spheres are feasible sources of beneficial microbes transferred from mother to neonate through delivery and feeding at the initial stage of life. Moreover, raw milk, especially from livestock, is easy to collect without serious ethical and technical concerns. In this context, several studies have reported the identification, isolation, and characterization of probiotic LAB from raw animal milk, such as buffalos [44,45,46], camels [47,48], cows [49,50,51], donkeys [52,53], goats [54,55,56,57,58,59], and water buffalos [60].

The Japanese-Saanen goat is a crossbreed of Japanese domestic and Saanen breeds, accounting for most dairy goats in Japan. Comprehensive milk composition analysis of Japanese-Saanen goats has been limited to date. Tomotake et al. [61] reported that Japanese-Saanen goat’s milk was characterized by higher amounts of saturated short-chain fatty acids, such as C_4:0_, C_6:0_, C_8:0_, and C_10:0_, than Holstein cow’s milk. The authors also demonstrated that the amount of α_S1_-casein in Japanese-Saanen goats was very low, probably due to the small number of α_S1_-casein alleles, giving less allergenicity than cow’s milk [61]. In addition, the microbiota in the milk of Japanese-Saanen goats remains unclear. In this study, we demonstrated the isolation, essential probiotic evaluation, and in vitro safety assessment of several LAB strains indigenous to Japanese-Saanen goat raw milk. To our knowledge, this is the first report of industrially potential probiotic LAB strains isolated from the raw milk of Japanese-Saanen goats.

## 2. Materials and Methods

### 2.1. Collection of Raw Milk from Goats

Raw milk was manually collected from five healthy female Japanese-Saanen goats in Tokachi Millennium Forest (42°55′56.5″ N 142°52′06.3″ E) in March 2014. In brief, the nipples were sanitized with 70% ethanol and predipped in nonoxynol-9 iodine solution before milking. Raw milk was collected using γ-ray-sterilized 50 mL conical tubes by a staff member wearing sterile rubber gloves in a goat barn. A certain amount of foremilk was discarded. The collected raw milk was immediately placed in a cooler box and transferred to the laboratory (approximately 30 km from the sampling site). The raw milk was then processed for microbial isolation within 1 h of collection.

### 2.2. Isolation of Bacteria from the Raw Milk

General chemicals used in this study were of analytical grade. Before isolation of LAB from the raw milk of goats, enrichment was performed. A 500 μL aliquot of each raw milk sample was inoculated aseptically into 5 mL of sterile de Man–Rogosa–Sharpe (MRS; Oxoid, Basingstoke, UK) broth and sterile M17 broth (Oxoid) supplemented with 2% glucose. The inoculated medium was incubated statically at 30, 35, and 42 °C for up to 72 h under aerobic and anaerobic conditions. To generate anaerobic atmosphere, Anaeropack Kenki (Mitsubishi Gas Chemical Co., Inc., Tokyo, Japan) was used. Each culture medium was serially diluted 10-fold (up to 10^8^ dilutions) in phosphate-buffered saline (PBS). A 100 μL aliquot of the diluted solution was smeared onto MRS and M17 agar plates containing 20 μg/mL cycloheximide and 1.1 mM bromocresol purple (BCP), and then incubated under the same conditions as above. Single colonies with yellow halos were picked from the agar plates, transferred into 5 mL of MRS or M17 broth, and incubated statically at the appropriate temperatures for 24 h under anaerobic conditions. To obtain pure colonies, single-colony isolation was performed for all isolates. The pure isolates were stored at −80 °C in 30% glycerol containing MRS until use.

### 2.3. Initial Characterization of the Isolated Bacteria

As an initial step for LAB isolation, bacteria that showed typical characteristics of LAB, such as Gram-positive rod- or sphere-shaped, catalase-negative, and gas- or non-gas-producer, were selected. First, a single colony of each bacterial strain was inoculated into MRS or M17 broth and then cultured under the same conditions as above. After incubation, a small portion of the culture medium was spotted on a glass slide and stained using a bacterial Gram-staining kit (Merck, Rahway, NJ, USA), according to the manufacturer’s instructions. Bacterial morphology and Gram reaction were observed under a light microscope. Second, a portion of hydrogen peroxide solution (FUJIFILM Wako Pure Chemical Co., Osaka, Japan) was added to the culture medium and spotted on a glass slide; formation of bubbles indicated catalase-positive bacteria. Third, a red-heated platinum inoculating loop was inserted into a similar culture medium. When bubble formation was observed, the bacteria were considered to be gas-producers.

### 2.4. Selection of Potential Probiotic Bacteria

In this study, tolerance to low pH and bile salts was used as the least index for the selection of potential probiotic bacteria, according to a previous report with slight modifications [62]. *Lacticaseibacillus rhamnosus* GG ATCC53103 (LGG) was purchased from the American Type Culture Collection, (Manassas, VA, USA) and used as a reference strain. Isolates that showed similar or higher tolerance to both low pH and bile salts than those of LGG were selected as potential probiotic bacteria. First, a single colony of each bacterium was inoculated into 5 mL of MRS or M17 broth and then precultured statically at an appropriate temperature for 48 h. A portion of the preculture medium was mixed with 5 mL of fresh MRS or M17 broth to achieve an optical density of 0.3 at OD600 nm, and then incubated at 37 °C for 18 h, corresponding to the end of the logarithmic growth phase. After cultivation, a 1 mL aliquot of the culture broth was collected and centrifuged at 15,000× *g* for 5 min, and the bacterial pellet was washed twice with PBS. The washed bacterial cells were incubated at 37 °C for 3 h in 5 mL of fresh MRS broth adjusted to pH 2.0, with 1 M HCl and 5 mL fresh MRS supplemented with 1.5% bile salts (Oxoid), either independently or sequentially. After that, the bacterial cells were harvested, washed as described above, resuspended in 5 mL fresh MRS, and incubated at 37 °C for 24 h. Survival rates of tested bacteria after low pH or bile salt treatment relative to that of LGG were estimated by comparing the OD600 nm values after 24 h of cultivation.

### 2.5. Identification of the Selected Potential Probiotic Bacteria Using 16S rDNA Sequencing

A single colony was selected and subjected to colony polymerase chain reaction (PCR) for 16S rDNA sequencing, according to a previous study [62]. Briefly, picked colonies were suspended in a 20 μL reaction mixture using ExTaq DNA polymerase (Takara Bio, Shiga, Japan), following the manufacturer’s instructions. The primers 16S_27F: 5′-AGAGTTTGATCCTGGCTCAG-3′ and 16S_520R: 5′-ACCGCGGCTGCTGGC-3′ were used to amplify the V1–V3 region of bacterial 16S rDNA. Amplicons were visualized using ethidium bromide staining under UV irradiation after 1.6% agarose gel electrophoresis, and then purified using the GenElute PCR clean-up kit (Merck KGaA, Darmstadt, Germany) according to the manufacturer’s instructions. Purified amplicons were stored at −20 °C until use. The sequencing reaction was performed using the purified amplicon as a template, 16S_27F as a primer, and the BigDye Terminator Cycle Sequencing Ready Reaction Mix Kit (Life Technologies Co., Carlsbad, CA, USA), following the manufacturer’s instructions. A homology search of the obtained rRNA sequence data was performed using the Basic Local Alignment Search Tool (BLAST) retrieval engine (https://blast.ncbi.nlm.nih.gov/Blast.cgi, accessed on 20 March 2015).

### 2.6. Evaluation of Antipathogenic Activities

Antipathogenic activities of the selected potential probiotic LAB were examined using disc diffusion assay, according to a previous report [63]. Stocks of safety level 2 pathogenic bacteria, *Salmonella enterica* subsp. *enterica* serovar. Typhimurium LT-2, *Shigella sonnei* strain No. 134, methicillin-resistant *Staphylococcus aureus* (MRSA) strain No. 29, methicillin-sensitive *Staphylococcus aureus* (MSSA) strain No. 18, *Listeria monocytogenes* No. 154, and *Escherichia coli* O157 strain No. S-12, all the clinical isolates of the Department of Veterinary Medicine, Obihiro University of Agriculture and Veterinary Medicine (OUAVM) were used. The propagation of pathogenic bacteria and the antipathogenic activity assay were performed at biosafety level 2 laboratories at OUAVM with permission from the safety committee of OUAVM. Briefly, cell-free culture supernatants (CFCSs) of the selected potential probiotic LAB were prepared. A 1 mL aliquot of 10^5^ to 10^6^ CFU/mL bacterial cell suspension was inoculated into 100 mL MRS and incubated at 37 °C for 24 h. Bacterial cells were then removed via centrifugation at 10,000× *g* at 4 °C for 30 min, and the supernatant was collected as CFCS. The CFCS was filter-sterilized using 0.2 μm filters (Advantec, Tokyo, Japan), and then lyophilized. Lyophilized CFCS was dissolved in sterilized 20 mM sodium phosphate buffer (pH 6.0) to obtain 20-fold concentrated CFCSs compared to the concentration before lyophilization. A 30 μL aliquot of the concentrated CFCS solution was added to Whatman no. 1 sterile disc filter papers with 6 mm diameter (GE Healthcare, Little Chalfont, UK) aseptically, air-dried at 25 °C for 1 h, and then placed on 1.2% (*w*/*v*) brain heart infusion (BHI, BD Biosciences, Spark, MD, USA) agar plates, containing 10^7^–10^9^ CFU/mL of pathogenic bacteria. The agar plates were then incubated at 4 °C for 1 h to diffuse CFCS into the agar and then incubated aerobically at 37 °C for 24 h. After incubation, antipathogenic activities were estimated via visual observation of clear zones around the disc filter papers. The antipathogenic activities of concentrated CFCSs neutralized to pH 7.0 with 1 M NaOH were also examined. Fresh sterilized 20 mM sodium phosphate buffer (pH 6.0) and 0.1 mg/mL ampicillin solution were used as the negative and positive controls, respectively.

### 2.7. Assessment of Antibiotic Resistance

Antibiotic resistance of the selected potential probiotic LAB was assessed, according to the ISO 10932/IDF 223 standard (2010) [64] and a previous report [63]. Minimal inhibitory concentrations (MICs) of ampicillin, chloramphenicol, clindamycin, erythromycin, gentamycin, kanamycin, streptomycin, and tetracycline (all antibiotics were purchased from Merck) were determined using the microdilution method using bacterial cells cultivated for 18 h as described above. Antibiotic resistance was evaluated by comparing to the MIC breakpoint values for *L. rhamnosus* recommended by the European Food Safety Authority Panel on Additives and Products or Substances used in Animal Feed [65].

### 2.8. Hydrolytic and Biotransformation Activities for Bile Acids

Hydrolysis and biotransformation of bile acids by the selected potential probiotic LAB were examined according to a previous report [63]. All bile acids used in this study, namely, taurocholic acid (TCA), taurochenodeoxycholic acid (TCDCA), taurodeoxycholic acid (TDCA), glycocholic acid (GCA), glycochenodeoxycholic acid (GCDCA), and glycodeoxycholic acid (GDCA), were purchased from Merck. To assess bile acids hydrolytic activities, bacterial cells were smeared on MRS agar plates containing 1 mM bile acids, and then incubated at 37 °C for 72 h. Hydrolytic activities were estimated via visual observation to check the emerged precipitation zone around colonies using *Enterococcus faecalis* ATCC 19433 as a positive control. To assess the biotransformation of cholic acid (CA) into deoxycholic acid (DCA), thin-layer chromatography (TLC) was performed. Bacterial cells were inoculated in 1/2MRS broth containing 0.15 mM sodium cholate and then incubated at 37 °C for 48 h anaerobically. After incubation, the pH of the broth was adjusted to 2.0 with 1 M HCl, and bile acids in the broth were extracted with 1 mL of ethyl acetate. The extracted bile acids were separated on a silica gel 60 TLC plate (Merck) using cyclohexane:ethyl acetate:acetic acid (7:23:3, *v*/*v*) as a developing solvent. Bile acids on the TLC plate were visualized by spraying 5% (*w*/*v*) phosphomolybdic acid in absolute ethanol, followed by heating until spots appeared.

### 2.9. Estimation of Enzyme Activities

To clarify any unfavorable enzymatic activity found in the selected potential probiotic LAB, an API ZYM kit (bioMérieux, Marcy I’Etoile, France) was used in this study, according to the manufacturer’s instructions. After 18 h of anaerobic incubation at 37 °C, bacterial cells were harvested, adjusted to 3 × 10^8^ CFU/mL with PBS, and applied to an API ZYM kit. Enzyme activity in the tested bacteria was evaluated using an API ZYM color chart (bioMérieux).

### 2.10. Evaluation of Mucin Degradation Activities

The ability of the selected potential probiotic LAB to degrade mucin was measured using two different methods, agar plate assay and test tube assay, according to a previous report with modifications [63]. Human fecal bacteria (HFB), used as a positive control in this study, were obtained from one of the coworkers under the regulations of the ethics committee on human-related studies at OUAVM. Autoclaved HFB (AHFB) was used as the negative control. Hog gastric mucin (HGM Type III; Merck) was purified and used in both experiments. Briefly, small circles of the bacterial cells were drawn on agar plates containing the following: 0.5% (*w*/*v*) HGM Type III, 7.5 g tryptone, 7.5 g casitone, 5.0 g yeast extract, 5.0 g beef extract, 5.0 g NaCl, 3.0 g K_2_HPO_4_·3H_2_O, 0.5 g KH_2_PO_4_, 0.5 g MgSO_4_·7H_2_O, 0.5 g L-cysteine HCl, 0.002 g resazurin, 15 g agarose, and 0 or 30 g glucose per liter of deionized water (pH 7.2). The agar plates were incubated under anaerobic conditions at 37 °C for 72 h. After incubation, agar plates were stained with 0.1% (*w*/*v*) amido black dissolved in 3.5 M acetic acid for 30 min at 25 °C, and then washed with 1.2 M acetic acid solution until clear zones were observed around colonies. Next, for the test tube assay, bacterial cells were incubated at 37 °C for 18 h in MRS, harvested, washed twice with PBS, inoculated in a medium with a similar composition to that used in the above assay, but without agar, and incubated at 37 °C for 48 h. After incubation, the remaining mucin in the cultivated medium was collected and subjected to 12% (*w*/*v*) sodium dodecyl sulfate-polyacrylamide gel electrophoresis (SDS-PAGE). After electrophoresis, gels were stained with Coomassie Brilliant Blue (CBB) and periodic acid–Schiff (PAS) reagents.

### 2.11. Assessment of Plasminogen Binding and Activation Abilities

The ability of the selected potential probiotic LAB to bind human plasminogen (hPlg, Merck) and hydrolyze hPlg was investigated according to a previous report [63]. Bacterial cells were cultivated in MRS for 24 h until the stationary phase, harvested, and washed twice with PBS. Then, bacterial cells were suspended in 40 μg/mL hPlg containing PBS at 10^9^ CFU/mL and incubated at 37 °C for 15 min. After incubation, the bacterial cells were harvested, washed twice with PBS, and resuspended in 250 μL of 50 mM Tris-HCl (pH 7.5). A 100 μL aliquot of the suspension was mixed with 30 μL of 0.54 mM D-valyl-leucyl-lysine-*p*-nitroanilide dihydrochloride (S-2251, Merck) and 0.24 kallikrein inhibitor unit (KIU) tissue plasminogen activator (tPA, Merck) or 0.06 KIU urokinase plasminogen activator (uPA, Merck), and then incubated at 37 °C for 1 h. Peptidic activity was monitored by measuring absorbance of the reaction mixture at the end point at a wavelength of 405 nm.

### 2.12. Hemolytic Activity Assay

The hemolytic activities of the selected potential probiotic LAB were examined using two different methods [63]. First, the activity was estimated using 5% sheep blood agar plates (Eiken Chemical Co., Ltd., Tokyo, Japan). Briefly, the population of bacterial cells, cultivated for 18 h as described above, was adjusted to 10^8^ colony forming units (CFU)/mL with PBS. A 10 μL aliquot of the bacterial cell suspension was smeared on a 5% sheep blood agar plate, and then incubated at 37 °C for 48 h under anaerobic conditions. Changes in the color surrounding the emerged colonies were visually observed. *Levilactobacillus brevis* ATCC 8287 was used as the negative control. Second, the activities were examined using a test tube assay. Red blood cells (RBCs) prepared from defibrinated sheep blood (Nippon Biotest Laboratory, Tokyo, Japan) were pelleted via centrifugation at 1500× *g* for 2 min at 25 °C and then washed three times with PBS. The 10^8^ RBC in 500 μL of PBS were mixed gently with 10^8^ CFU of bacterial cells in 500 μL of PBS. A total of 500 μL of the mixture was incubated at 37 °C for 1.5 h, centrifuged at 1500× *g* for 10 min at 25 °C, and then the absorbance of the supernatant was measured at a wavelength of 405 nm. Supernatants obtained when RBCs were mixed with 1% (*v*/*v*) Triton X-100 (Merck) and PBS were used as the positive and negative controls, respectively.

### 2.13. Statistical Analysis

Numerical data, except the antibiotic susceptibility test, were expressed as means ± standard deviation (SD) from three independent experiments. The statistical significance was assessed by Student’s *t*-test for low pH and bile salt tolerance assay or one-way analysis of variance (ANOVA) with Tukey’s post hoc test for antimicrobial and plasminogen binding/activation assays, using XLSTAT 4.03 software (Social Survey Research Information Co., Ltd., Tokyo, Japan). Data were considered significant at a *p* value less than 0.05.

## 3. Results

### 3.1. Isolation and Identification of Potential Probiotic LAB

A total of 101 strains were obtained from enrichment cultures of raw milk from five healthy female Japanese-Saanen goats. All strains were identified as Gram-positive bacteria and harbored catalase-negative and no gas-producing characteristics (Table 1). Microscopic observations revealed five different bacterial morphologies, indicating 5 monococci, 31 diplococci, 24 tetracocci, and 41 streptococci or bacilli. Of these, 41 strains showed similar or higher tolerance to either low pH or bile salts compared to LGG (Figure 1A). Several strains isolated in this study showed apparently higher tolerance to low pH than LGG, but most of them were less tolerant to bile salts than LGG. Only two strains showed tolerance to both low pH and bile salts that are comparable to LGG. These two strains, named as YM2-1 and YM2-3 (see Figure 1A), were isolated under anaerobic MRS culture conditions at 42 °C. These two strains were further examined for their tolerance to successive treatments with low pH and bile salts. Both strains showed tolerance to low pH and bile salts approximately 60% lower than that of LGG (Figure 1B). Light microscopic images of YM2-1 and Y2-3 are shown in Figure 1C. BLAST similarity search revealed that both YM2-1 and YM2-3 were closely related to *L. rhamnosus* LC3 (query coverage 98%, sequence identity 99%) and *L. rhamnosus* 330 (query coverage 97%, sequence identity 99%), respectively. Finally, *L. rhamnosus* YM2-1 and YM2-3 were selected as probiotic LAB candidates for further investigation.

### 3.2. The Antipathogenic Activities of L. Rhamnosus YM2-1 and YM2-3

Consequently, concentrated CFCSs of both strains showed antipathogenic activities to a similar extent against tested pathogenic bacteria, except for *S. enterica* Typimurium (Figure 2 and Table 2). The concentrated CFCS of strain YM2-3 showed significantly higher antipathogenic activity than that of YM2-1, regardless of neutralization treatment. The neutralization of concentrated CFCSs significantly decreased and abolished antipathogenic activity against *S. sonnei* and *L. monocytogenes*, respectively. *S. enterica* Typimurium, MRSA, and *E. coli* O157 showed resistance against 0.1 mg/mL ampicillin. Two different types of inhibition zones were identified by visual observation of the BHI agar plates. One is low transparency with relatively large diameters, which were observed for *S. enterica* Typhimurium, *S. sonnei*, and *L. monocytogenes*, and the other is high transparency with small diameters, observed for MRSA, MSSA, and *E. coli* O157. With the exception of *L. monocytogenes*, antipathogenic activities in concentrated CFCSs of both strains remained after neutralization by addition of 1 M NaOH (Figure 2).

### 3.3. Safety Assessments of L. Rhamnosus YM2-1 and YM2-3

To ensure safe use of *L. rhamnosus* YM2-1 and YM2-3 in food industries, several preliminary in vitro experiments were performed in this study. First, to effectively control unfavorable propagation of the two strains, their antibiotic susceptibilities were examined. Consequently, the two strains showed identical antibiotic profiles (Table 3). Among tested antibiotics, both strains were susceptible to ampicillin (MIC = 0.25 µg/mL), gentamycin (MIC = 4 µg/mL), kanamycin (MIC = 32 µg/mL), streptomycin (MIC = 16 µg/mL), tetracycline (MIC = 4 µg/mL), and chloramphenicol (MIC = 4 µg/mL), whereas they were not susceptible to clindamycin (MIC = 16 µg/mL) and erythromycin (MIC = 4 µg/mL), according to the MIC breakpoint values for *L. rhamnosus* recommended by EFSA [65]. The same MIC values were obtained for three independent experiments. Comparison of the MIC values of the two strains and LGG revealed differences when gentamycin and tetracycline were tested. LGG showed two- and four-fold smaller MIC values against gentamycin and tetracycline, respectively, than the other two strains. Therefore, the two strains were slightly more resistant to gentamycin and tetracycline than to LGG. Accordingly, ampicillin was the most effective antibiotic against *L. rhamnosus* YM2-1 and YM2-3.

Bioconversion of bile acids and certain enzymes secreted by the intestinal microbiota have been suggested to cause carcinogenicity and inflammation in the intestinal rumen of host animals [66,67]. In this regard, the ability of *L. rhamnosus* YM2-1 and YM2-3 to hydrolyze six typical bile acids, TCA, TCDCA, TDCA, GCA, GCDCA, and GDCA; convert a representative primary bile acid (CA) to suspected carcinogenic secondary bile acid (DCA); and clarify the activities of enzymes, including carcinogenesis-related hydrolases, such as α-galactosidase, β-glucuronidase, and *N*-acetyl-β-glucosaminidase, was examined. Results showed no precipitation around the colonies of *L. rhamnosus* YM2-1 and YM2-3 on all agar plates containing each of the six bile acids, indicating that both strains have no hydrolytic activity on such typical bile acids (Figure 3A,B). TLC analysis confirmed that no bioconversion from primary bile acid (CA) to secondary bile acid (DCA) was caused by the action of the two strains (Figure 3C). Enzyme activities, especially carcinogenesis-related hydrolases, such as α-galactosidase (E13 in Table 4), β-glucuronidase (E15 in Table 4), and *N*-acetyl-β-glucosaminidase (E18 in Table 4) of both strains, as well as LGG, were lower than the detection limits of the API ZYME kit.

Mucin is a physical barrier in the intestinal epithelium of the host; hence, many commensal and pathogenic bacteria can bind to and degrade mucin. In fact, several commensal bacteria, such as *Bifidobacterium* species, are capable of degrading mucin to utilize it as an energy source [68], but the activity is not strong enough to collapse the barrier function of mucin. Nevertheless, to avoid any changes in mucus content and structure that compromise the barrier function of the mucus layer [69], the mucin degradation activities of the two strains were examined. Consequently, no apparent clear zone was observed around the colonies of the two strains on 0.5% (*w*/*v*) HGM Type III agar plates, indicating that neither strain had mucin-degrading activity (Figure 4A). This result was further supported using SDS-PAGE analysis, wherein no smearing of mucin bands was observed for the two strains in contrast to the positive control, HFB (Figure 4B). Hog gastric mucin has been reported to be structurally and chemically similar to human gastric mucin [70]. Therefore, *L. rhamnosus* YM2-1 and YM2-3 may also be unable to degrade human gastric mucin.

Plasminogen is a proenzyme that not only circulates in the blood stream and plays a significant role in fibrinolysis but is also present in the extracellular matrix (ECM), which is a complex of several glycoproteins, in association with its reconstitution [71]. Plasminogens need to be activated by tPA or uPA to exhibit protease activity similar to plasmin. Pathogenic bacteria are known to bind host-derived plasminogen in the gastrointestinal tract, activate plasminogen, and utilize it as a tool to degrade the host ECM [71]. Therefore, probiotics should ideally not possess plasminogen-activating activity. In the present study, no statistically significant difference was observed in the hydrolysis of the chromogenic plasmin substrate, S-2251, between the negative controls and YM2-1 or YM2-3 (Figure 5). Therefore, it was concluded that both strains were unable to hydrolyze plasminogen in the presence of tPA and uPA into its active form, plasmin.

Finally, the hemolytic activities of *L. rhamnosus* YM2-1 and YM2-3 were examined. Hemolysis can be triggered by the presence of hydrogen peroxide, which is commonly produced by LAB as an antimicrobial substance [72], and hemolysins, which are lipids and proteins that are capable of disturbing the cell membrane of erythrocytes [73]. As a consequence, strains YM2-1 and YM2-3 showed slight changes in the color of 5% sheep blood agar under the colonies, but no color change was observed in their surrounding areas, similar to LGG (Figure 6A). However, after incubation with RBCs, no hemoglobin leakage was observed in either strain, similar to PBS used as a negative control, in contrast to 1% (*v*/*v*) Triton X-100 used as a positive control (Figure 6B). Therefore, it was concluded that YM2-1 and YM2-3 had no hemolytic activity.

## 4. Discussion

Predominant bacteria in the raw milk of goats generally belong to the phylum Proteobacteria. Phylum Firmicutes can also be found, but the population is not relatively high [74,75]. In recent studies, major LAB species isolated as probiotics from the raw milk of goats were found to belong to the genera *Enterococcus*, *Lactobacillus*, *Lactococcus*, *Leuconostoc*, and *Streptococcus* [54,56,57,76,77,78,79,80,81,82]. Hence, the cocci found as major isolates in this study are likely to be members of *Enterococcus*, *Lactococcus*, *Leuconostoc*, and *Streptococcus*. Similar to our study, candidate probiotic *L. rhamnosus* has been isolated from Algerian goat milk [81,82]. As reported by Marroki et al. [82], the lactobacilli most commonly found in goat milk are *L. plantarum*, *L. rhamnosus*, *L. casei*, and *L. paracasei*, implying that raw milk from goat is a promising source of industrially important LAB. Delavenne et al. [83] reported seasonal changes in the ability of raw milk as a productive reservoir for antifungal lactobacilli. Characteristics related to the manufacture of fermented milk products, such as proteolytic, lipolytic, and exopolysaccharide production, are yet to be elucidated for *L. rhamnosus* YM2-1 and YM2-3. Therefore, such challenges are future concerns to declare that these two strains are suitable starters for milk fermentation.

Antimicrobial activities, especially against pathogenic bacteria, are well-established characteristics of LAB. One reason for the antimicrobial activity of LAB is their high potency to compete for nutrients with other microbes. Another is their ability to produce antimicrobial metabolites, such as organic acids, hydrogen peroxide, bacteriocins, and antifungal peptides [84]. Among the antimicrobial metabolites of LAB, bacteriocins are fascinating in terms of their application as safe biopreservatives in the food industry. Indeed, several bacteriocin-producing LAB have been isolated from the raw milk of goats, such as *Enterococcus hirae* [85,86], *Enterococcus mundtii* [56], *Lactococcus lactis* [87], and *Pediococcus acidilactici* [88]. Some antipathogenic activities of CFCSs from *L. rhamnosus* YM2-1 and YM2-3 persisted after neutralization. This observation indicated the presence of antipathogenic substances, such as bacteriocins other than organic acids, in the CFCSs, although further investigations are required. Finally, it was concluded that *L. rhamnosus* YM2-1 and YM2-3 harbored probiotic characteristics and antimicrobial activities against pathogenic bacteria.

Clindamycin and erythromycin are categorized as macrolides and lincosamides, respectively, and are structurally diverse. However, they show similar antibiotic mechanisms because of their overlapping binding sites in the peptidyl transferase region in the 23S rRNA of bacteria [89]. In general, bacterial resistance to antibiotics occurs through the acquisition of antibiotic-resistance genes in plasmids or genomes via lateral gene transfer [90,91]. Genes responsible for bacterial resistance to macrolide, lincosamide, streptogramin, ketolide, and oxazolidinone have been reported to include *ere*, *erm*, *mef*, *mph*, and *msr* [92]. The existence of *erm*(A), *erm*(B), and *erm*(C), which encode methyltransferases that modify the antibiotic-binding site in 23S rRNA, has been confirmed in erythromycin-resistant *L. rhamnosus* Pen [93]. Mutations in ribosomal proteins are known to evoke bacterial acquisition of resistance against macrolide and lincosamide antibiotics [94]. Recently, Biswas et al. [95] found a macrolide-resistant *L. rhamnosus* mutant, in which 75 base pairs were deleted from the *rplD* gene, which encodes the large ribosomal subunit L4. Although the molecular mechanism is not clear, it is noteworthy that mutations in ribosomal proteins should also be considered in the investigation of antibiotic-resistance acquisition in *L. rhamnosus* species.

The disadvantages of bile salt hydrolytic activities in LAB are controversial, as Pisano et al. [96] reported. It is assumed that bile salt hydrolytic activity of *Lactobacillus* species itself is not necessarily harmful, because they generally lack dehydroxylating activities for deconjugated bile salts, which bind to bile acid receptors and evoke a proinflammatory response in the colon [67,97]. More curiously, recent studies have revealed the preferable role of secondary bile acids. The bile acid 7α-dehydroxylating gut bacterium *Clostridium scindens* is capable of biotransforming primary bile acids into secondary bile acids, such as DCA and lithocholic acid. Furthermore, *C. scindens* secretes a tryptophan-derived antibiotic, 1-acetyl-β-carboline, which inhibits in vivo growth of *Clostridium difficile*, and the inhibition activity is enhanced by secondary bile acids [98]. On the other hand, *Lactobacillus johnsonii* La1 could inhibit the growth of pathogenic protozoa *Giardia duodenalis* in vitro, due to its ability to deconjugate primary bile acids produced by the action of extracellular enzymes [99]. Therefore, the bile acid bioconversion activities of LAB should be carefully considered in terms of their health benefits for human beings. β-Glucosidases of intestinal bacteria are assumed to be involved in the hydrolysis of glycoconjugates, such as plant-derived secondary metabolites that release either toxic or health-beneficial aglycones [100]. It is unclear how much more impact will be given by the lower β-glucosidase activities in strains YM2-1 and YM2-3 than in LGG.

Several pathogenic bacteria can immobilize the host’s plasminogen on the bacterial cell surface via plasminogen-binding receptors, and then process the plasminogen using either the host’s tPA, uPA, or endogenous enzymes, such as streptokinases and staphylokinases [101]. Plasminogen-binding receptors have not only been found in pathogenic bacteria, but also in commensal bacteria. For example, moonlighting α-enolase was found to be a plasminogen-binding receptor on the bacterial cell surface of *L. plantarum* LM3 [101]. A fascinating idea is that such harmless commensal bacteria capable of binding plasminogen but incapable of activating it might competitively interfere with pathogenic bacteria in plasminogen-mediated invasion of the host [101]. Overexpression of the uPA/uPA receptor system has been found to be incorporated into the molecular mechanism of several cancer invasions. For example, the protein p185 encoded by *c-erbB-2* was suggested to be a promoter of invasion and metastasis of gastric cancer, and its expression level was strongly correlated with higher expression levels of tumor-associated proteases and inhibitors, including the uPA/uPA receptor system [102]. Rasouli et al. [103] demonstrated that *Lactobacillus reuteri* (PTCC 1655) might prevent gastric cancer progression by downregulating the expression of uPA and uPA receptor genes in vitro. Therefore, only binding to plasminogen or plasminogen activators could be a beneficial characteristic of probiotic LAB.

## 5. Conclusions

In conclusion, potential probiotic strains *L. rhamnosus* YM2-1 and YM2-3 were successfully isolated from the raw milk of Japanese-Saanen goats. These strains showed antimicrobial activities comparable to those of LGG, a well-established commercial probiotic strain. The results of all the in vitro safety evaluations confirmed their safe use in the food industry, such as in cheese manufacturing. For further safety evaluations in vitro, the production abilities of biogenic amines and DNase and gelatinase activity can be considered. Additional studies investigating heath-beneficial characteristics such as antidiabetic, antitumor, anti-inflammatory, hypotensive, and immunomodulating activities are required to provide further insights into the two strains.

## Figures and Tables

**Figure 1 animals-13-00007-f001:**
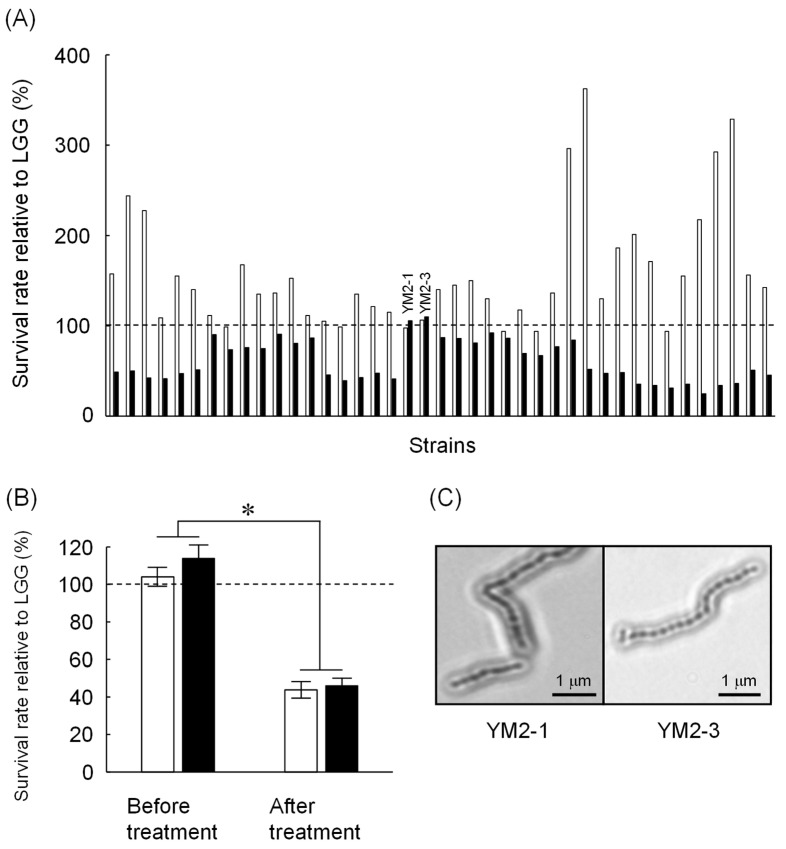
Screening of potential probiotic LAB isolated from Japanese-Saanen goat raw milk. (**A**) Tolerance of isolated LAB strains against low pH (blank bars) and 1.5% bile salts (filled bars). Survival rates of the tested strains relative to LGG are indicated. Based on comparable bile salts tolerance to LGG, YM2-1 and YM2-3 strains were selected for further experiments. (**B**) Tolerance of YM2-1 and YM2-3 against successive treatment of low pH and 1.5% bile salts (*n* = 3). Survival rates of the tested strains relative to LGG are shown. YM2-1 and YM2-3 were less tolerant against the treatment compared to LGG. Data were analyzed statistically by Student’s *t*-test. Significant difference is indicated by asterisk. (**C**) Microscopic observation of YM2-1 and YM2-3 strains after Gram staining.

**Figure 2 animals-13-00007-f002:**
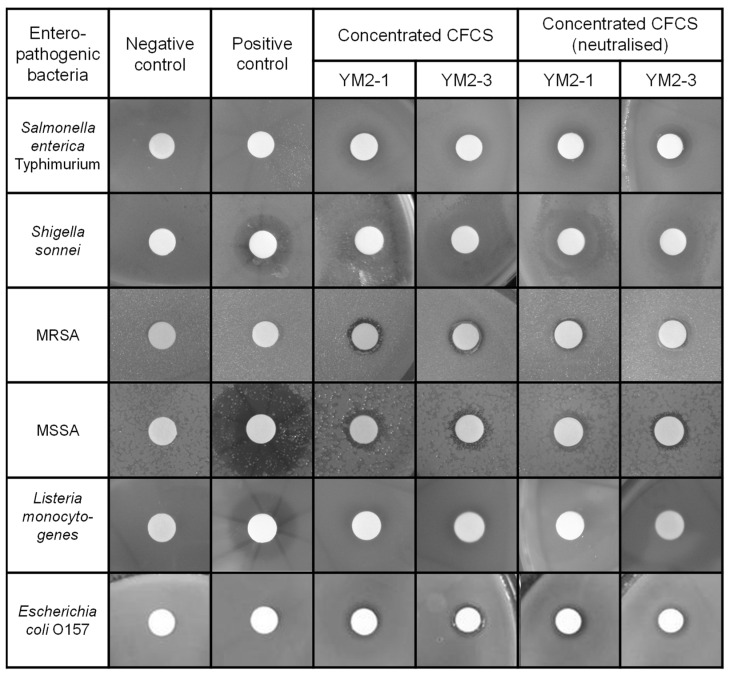
Antipathogenic activities of concentrated CFCS prepared from culture broths of YM2-1 and YM2-3 strains. Antipathogenic activities against *Salmonella enterica* Typhimurium, *Shigella sonnei*, methicillin-resistant *Staphylococcus aureus*, methicillin-sensitive *Staphylococcus aureus*, *Listeria monocytogenes*, and *Escherichia coli* O157 were examined by observing the inhibition zone formation around the filter disc, in which concentrated CFCSs were applied. To confirm presence of antipathogenic substances in the concentrated CFCS, neutralized concentrated CFCSs were also examined. Representative images are shown. Negative control, fresh sterilized 20 mM sodium phosphate buffer (pH 6.0). Positive control, 0.1 mg/mL ampicillin.

**Figure 3 animals-13-00007-f003:**
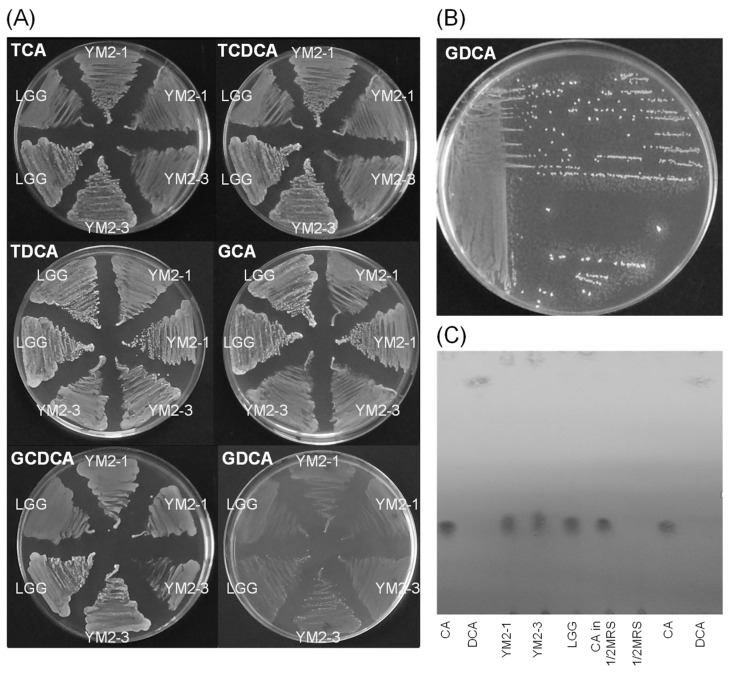
Effects of YM2-1 and YM2-3 strains on bile acid biotransformation. (**A**) Bile acid hydrolyzing activities were estimated for YM2-1 and YM2-3 strains, using LGG as a reference strain by visual observation of the precipitation zone, which consisted of insoluble calcium salt of bile acid, around bacterial colonies on 1 mM bile acids containing MRS agar plates. (**B**) Positive control, *Enterococcus faecalis* ATCC 19433. Tiny white dots around bacterial colonies indicate precipitates of a hydrolyzed bile acid, GDCA. (**C**) Bioconversion activities of YM2-1 and YM2-3 strains from CA, a primary bile acid, to DCA, a secondary bile acid. No bioconversion activity from CA to DCA was observed for either strain.

**Figure 4 animals-13-00007-f004:**
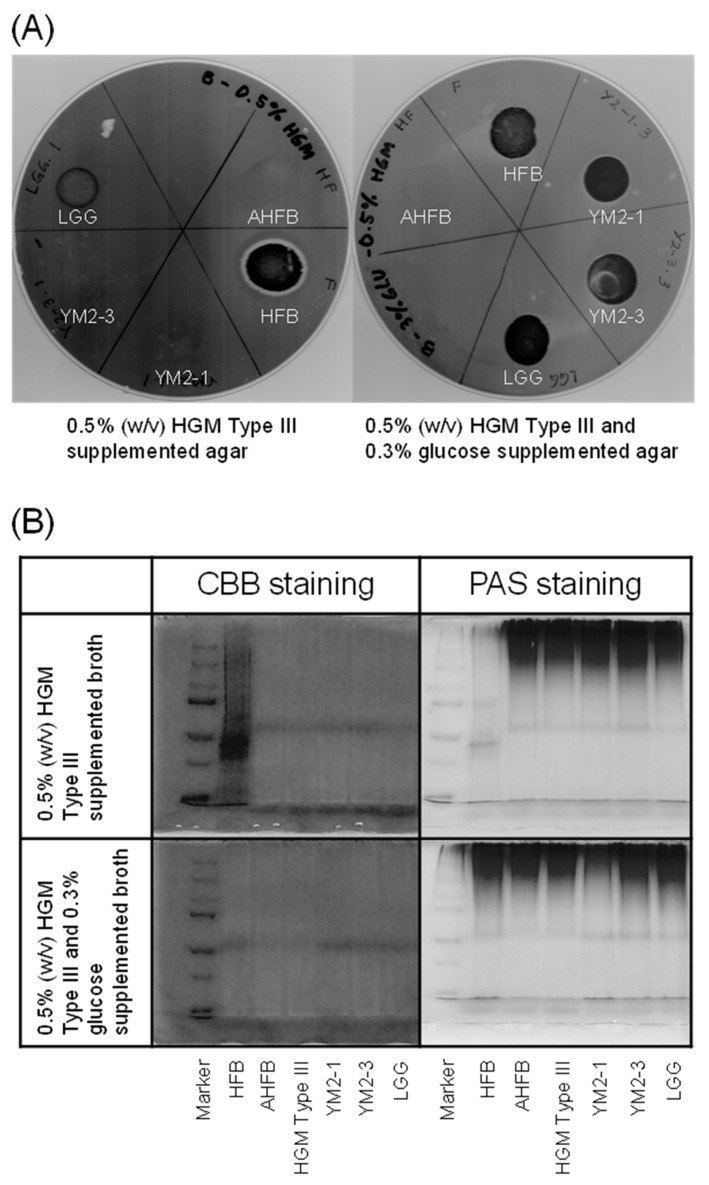
Mucin degradation activities of YM2-1 and YM2-3 strains. (**A**) 0.5% (*w*/*v*) HGM Type III containing agar plates stained with 0.1% (*w*/*v*) amido black. Clear zone represents mucin degradation activity of the tested bacteria. (**B**) Evaluation of mucin degradation activity in vitro on SDS-PAGE gels stained with CBB and PAS reagents. Loss of high-molecular-weight band and presence of smear band in lower-molecular-weight range indicate mucin degradation activity. Marker, precision plus protein dual color standards (Bio-Rad Laboratories, Hercules, CA, USA); HFB, human fecal bacteria (positive control); AHFB, autoclaved human fecal bacteria (negative control); HGM Type III (reference).

**Figure 5 animals-13-00007-f005:**
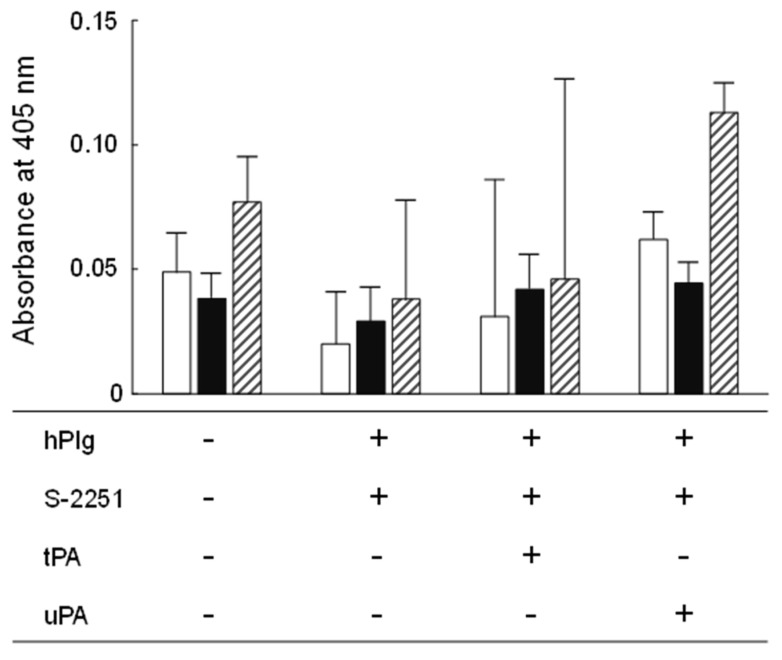
Plasminogen binding and activation activities of YM2-1 and YM2-3 strains. The experiment was performed in three independent trials, and then obtained data were statistically analyzed using one-way ANOVA with Tukey’s post hoc test. Data were considered significant at a *p* value less than 0.05. Consequently, no significant difference was found in the activities among YM2-1, YM2-3, and LGG under different experimental conditions. hPlg, human plasminogen; S-2251, 0.54 mM D-valyl-leucyl-lysine-*p*-nitroanilide dihydrochloride; tPA, 0.24 KIU tissue plasminogen activator; uPA, 0.06 KIU urokinase plasminogen activator.

**Figure 6 animals-13-00007-f006:**
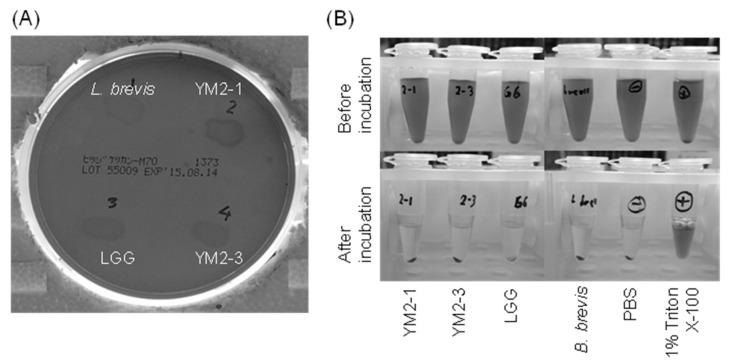
Hemolytic activities of YM2-1 and YM2-3 strains investigated (**A**) on 5% sheep blood agar plates and (**B**) in vitro against RBCs prepared from defibrinated sheep blood. The 1% (*v*/*v*) Triton X-100 was used as a positive control. *Levilactobacillus brevis* ATCC 8287 and PBS were used as negative controls.

**Table 1 animals-13-00007-t001:** Summary of culture conditions and general characteristics of isolated bacteria from raw goats milk.

Medium	Aerobicity	Incubation Temperature (°C)	Gram Staining	Catalase Activity	Gas Production	Bacterial Cell Morphology	Number of Isolated Strains
MRS	Anaerobic	42	Positive	Negative	Negative	Diplococci	7
						Tetracocci	4
						Streptococci or Bacilli	27
		35	Positive	Negative	Negative	Monococci	3
						Diplococci	3
						Tetracocci	11
						Streptococci or Bacilli	14
		30	Positive	Negative	Negative	Monococci	1
						Diplococci	21
						Tetracocci	2
M17	Anaerobic	42	Positive	Negative	Negative	Monococci	1
	Aerobic	30	Positive	Negative	Negative	Tetracocci	7
							101

Results of the culture conditions in which colonies emerged are indicated.

**Table 2 animals-13-00007-t002:** Antimicrobial activity of concentrated CFCSs of YM2-1 and YM2-3 against six pathogenic bacteria.

Pathogenic Bacteria	Antimicrobial Activity (Diameter in mm)				
	YM2-1		YM2-3		0.1 mg/mL Ampicillin
	Before Neutralization	After Neutralization	Before Neutralization	After Neutralization	
*S. enterica* Typhimurium	0.71 ± 0.01 ^aC^	0.76 ± 0.01 ^aC^	1.07 ± 0.06 ^AC^	0.98 ± 0.12 ^AC^	0 ^c^
*S. sonnei*	2.0 ± 0.11 ^BC^	1.3 ± 0.27 ^b^	1.8 ± 0.18 ^BC^	1.1 ± 0.03 ^b^	1.2 ± 0.02 ^c^
MRSA	0.83 ± 0.02 ^C^	0.76 ± 0.01 ^C^	0.76 ± 0.07 ^C^	0.69 ± 0.01 ^C^	0 ^c^
MSSA	0.83 ± 0.06 ^c^	0.75 ± 0.07 ^c^	0.80 ± 0.03 ^c^	0.78 ± 0.03 ^c^	2.1 ± 0.04 ^C^
*L. monocytogenes*	0.87 ± 0.01 ^Bc^	0 ^bc^	0.88 ± 0.02 ^Bc^	0 ^bc^	1.4 ± 0.02 ^C^
*E. coli* O157	0.68 ± 0.01 ^C^	0.69 ± 0.01 ^C^	0.72 ± 0.04 ^C^	0.71 ± 0.03 ^C^	0 ^c^

The antimicrobial activities were evaluated by measuring diameters of growth inhibition zones. The data are expressed as mean ± SD and were statistically analyzed using one-way ANOVA with Tukey’s post hoc test. *p* < 0.05 (*n* = 3); ^A,a^ Significant difference between YM2-1 and YM2-3; ^B,b^ Significant difference between before and after neutralization; ^C,c^ Significant difference between the strains and 0.1 mg/mL ampicillin.

**Table 3 animals-13-00007-t003:** Antibiotic susceptibilities of *L. rhamnosus* YM2-1, Y2-3, and GG.

Strain	MIC Values against Antibiotics (μg/mL)							
	Am	Cm	Cl	Em	Gm	Km	Sm	Tc
LGG	0.25	4	16	8	2	64	16	1
YM2-1	0.25	4	16	8	4	64	16	4
YM2-3	0.25	4	16	8	4	64	16	4
MIC BP	4	4	1	1	16	64	32	4

Am, ampicillin; Cm, chloramphenicol; Cl, clindamycin; Em, erythromycin; Gm, gentamycin; Km, kanamycin; Sm, streptomycin; Tc, tetracycline; MIC BP, minimal inhibitory concentration break point recommended by EFSA (2012) [65].

**Table 4 animals-13-00007-t004:** Enzymatic activities of YM2-1 and YM2-3 strains.

Strains	Enzyme Activities *																			
	E1	E2	E3	E4	E5	E6	E7	E8	E9	E10	E11	E12	E13	E14	E15	E16	E17	E18	E19	E20
LGG	+	+	+	+	−	+++	+++	++	−	+	++	+++	−	+++	−	+	+++	−	−	++
YM2-1	+	+	+	+	−	+++	+++	+	−	+	+	+++	−	++	−	+	+	−	−	+
YM2-3	+	+	+	+	−	+++	+++	+	−	+	++	+++	−	++	−	+	+	−	−	+

* E1, control (no substrate); E2, alkaline phosphatase; E3, esterase (C4); E4, esterase/lipase (C8); E5, lipase (C14); E6, leucine arylamidase; E7, valine arylamidase; E8, cystine arylamidase; E9, trypsin; E10, α-chymotrypsin; E11, acid phosphatase; E12, naphthol-AS-BI-phosphohydrolase; E13, α-galactosidase; E14, β-galactosidase; E15, β-glucuronidase; E16, α-glucosidase; E17, β-glucosidase; E18, N-acetyl-β-glucosaminidase; E19, α-mannosidase; E20, α-fucosidase.

## Data Availability

The data presented in this study are available on request from the corresponding author.

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
