# Peer review of "In Vitro Probiotic Characterization and Safety Assessment of Lactic Acid Bacteria Isolated from Raw Milk of Japanese-Saanen Goat (Capra hircus)"

_animals, 2022, doi:10.3390/ani13010007_

Round 1

Reviewer 1 Report

Point 1: Overall, the study is well designed, and clearly presented. However, some sections need to be developped and enriched.
Although the flaws within the manuscript, I suggest its publication in case of revision. Some indications for major revisions are given below.

Point 2: Have the 16S rRNA sequence of the L. rhamnosus YM2-1 and YM2-3 strains been deposited in the publicly available database?

Point 3: Why you did not search for genes coding for bacteriocins?

Point 4: Why you were limited to a limited number of tests in order to assess the probiotic aspects of the bacterial strains? I propose an autoaggregation and coaggregation study.

Point: Describe the potential use in the food industry for the two tested strains of L. rhamnosus YM2-1 and YM2-3.

Author Response

Point 1: Overall, the study is well designed, and clearly presented. However, some sections need to be developped and enriched.

Although the flaws within the manuscript, I suggest its publication in case of revision.

Some indications for major revisions are given below.

[AU] The authors thank you for reviewing our manuscript. Your valuable comments and suggestions are highly appreciated.

Point 2: Have the 16S rRNA sequence of the L. rhamnosus YM2-1 and YM2-3 strains been deposited in the publicly available database?

[AU] No, not yet. However, we are currently doing whole genome sequencing of these strains, and therefore after it finished, we will deposit the genome data into GenBank.

Point 3: Why you did not search for genes coding for bacteriocins?

[AU] As the same reason above, we did not try it in this study.

Point 4: Why you were limited to a limited number of tests in order to assess the probiotic aspects of the bacterial strains? I propose an autoaggregation and coaggregation study.

[AU] We totally agree that autoaggregation and coaggregation are important properties to be examined. However, it is practically difficult to include such experimental data in the current version of the manuscript, mainly due to the limited time frame. Therefore, we will perform the thorough probiotic characterization of the strains, including autoaggregation and coaggregation, in the next study.

Point: Describe the potential use in the food industry for the two tested strains of L. rhamnosus YM2-1 and YM2-3.

[AU] One possibility is that application of these strains as adjunct starters for cheese manufacturing. In fact, we have tried to make affine cheese using goat milk and it was successful. It is not published data, but cheese texture was influenced by the adjunct starter, so that we are currently doing some investigation on it. Accordingly, “such as cheese manufacturing” was inserted in line 548 in the conclusion section.

Reviewer 2 Report

The manuscript by Yukimune Tanaka et al. isolated and investigated the safety of two lactic acid bacteria. This study is well-organized and this study is of interest to the readers. I have the following suggestions and comments:

1, the authors must double-check the English of the manuscript. Some of the species name should be italic. For example, L. rhamnosus (line 20),  Levilactobacillus brevis (line 269) etc. 

2, In Figure 1A, Figure 3D, the error bars should be added. The experiments should be repeated. 

3, For the antibiotic resistance experiments, the authors should further study the genomes of the two bacteria and check the antibiotic resistance genes. 

4, For the safety studies of the two bacteria, the authors should investigate the safty using animal experiments. Generally, this is a requirement. 

Author Response

The manuscript by Yukimune Tanaka et al. isolated and investigated the safety of two lactic acid bacteria. This study is well-organized and this study is of interest to the readers. I have the following suggestions and comments:
[AU] The authors thank you for reviewing our manuscript. Your valuable comments and suggestions are highly appreciated.

1, the authors must double-check the English of the manuscript. Some of the species name should be italic. For example, L. rhamnosus (line 20),  Levilactobacillus brevis (line 269) etc. 
[AU] English proof reading has been performed by a commercial editing service, Editage. Typos have been checked again by the authors throughout the manuscript. Style of bacterial species names were also checked throughout the manuscript.

2, In Figure 1A, Figure 3D, the error bars should be added. The experiments should be repeated.
[AU] For Figure 1A, it is practically impossible to do the experiment again, because some of the isolates have already been lost. The purpose of this experiment is screening of probiotic candidates which indicate comparable or higher tolerance against low pH and bile salts than those of LGG. Without statistical data, we might miss several more candidates, which indicated slightly lower bile salt tolerance as shown in Figure 1A. We believe that this gives no critical influence on the conclusion of this manuscript. We realize that APIZYME assay is not quantitative experiment which needs statistical analysis, so that it is not suitable to indicate the result in a bar graph representation. Therefore, this result is now shown in a new Table 4. Accordingly, the sentence “It should be noted that b-glucosidase (E17 in Fig. 3D) activity in YM2-1 and YM2-3 was several-fold lower than that of LGG.” in lines 350-351 is deleted.

3, For the antibiotic resistance experiments, the authors should further study the genomes of the two bacteria and check the antibiotic resistance genes.

[AU] Thank you for the suggestion. This manuscript is aiming to report the first screening of potential probiotic strains isolated from raw milk of Japanese-Saanen goat. We choose antipathogenic activity as a probiotic characteristic in this study. Therefore, we consider genomic studies in relation to the antibiotic resistance of the isolated strains as our next challenge.

4, For the safety studies of the two bacteria, the authors should investigate the safty using animal experiments. Generally, this is a requirement. 

[AU] As the reviewer pointed out, animal experiment is in general prerequisite for safety assessment of probiotics especially for commercial use. However, there are several reports exist in which only in vitro safety assessment for probiotic candidate strains was described. Therefore, we decided to add the term “in vitro” in the title. So, the new title is “In vitro Probiotic Characterisation and Safety Assessment of Lactic Acid Bacteria Isolated from Raw Milk of Japanese-Saanen Goat (Capra hircus).

Reviewer 3 Report

Reviewer's Comment:

The manuscript describes the probiotic characterization and the safety assessment of Lactic Acid bacteria isolated from raw milk of Japanese-Saanen Goat (Capra hircus). The paper is very interesting, but it must be improved. Is necessary to repeat all the tests at least 3 times to bee scientifically valid.

    Simple Summary

Line 19: Staficolococci are not enteropathogenic bacteria. I suggest changing the terms in antibacterial activity.

Line 20: “L. rhamnosus” must be written in italics

  Abstract

Line 30: see the comment reported for line 19.

Keywords: “anti-enteropthogenic activity” see the comment reported for line 19

  Introduction Lines 66-70: Not a clear sentence, please re-write it.  

Materials and Methods

Line 107: “with 2% glucose” insert data on the glucose used (Producer, Town, State), here and for all the media/substance and reagent used.

Line 108. “in anaerobic conditions” insert data on the anaerobic conditioner.

Line 109, 111, etc… : See the comment reported in line 107 “phosphate-buffered saline (PBS)”,”cycloheximide”, bromocresol purple (BCP), ect… please check in all the paper. (Line 299,300, 304, etc.…)

Lines 114-115: no clear procedure, please explain better.

Line 116: “..in 30% glycerol until use” insert the media used as 70%.

Line 133: “Lacticaseibacillus rhamnosus” change in italics

Lines 164-191: see the comment reported for line 19.

Lines 166-170: The used strain are wild strains or certified. Please insert some other information.

Line 175: “105 to 106 CFU/mL” please verify

Lines 185-186: “The agar plates were then incubated at 4°C for 1 h…” please explain the reason of this passage.

Lines 164-191: Is not reported the number of repetitions performed. In the

Lines 193-277: for all the experiments performed are not reported the repetition (at least 3 must be performed). So, I think that they have performed only one time, therefor the results reported are not significant.

Results

Table 1: is an image with low quality. Please insert a table not as a picture. The description of the procedure reported in the Material and methods is not equal to those reported in the Table. In the Table 1, no anaerobic condition is reported only for MRS; and for M17 is reported only 1 temperature 42°C in anaerobic and 30°C in aerobic. Please explain better or modify the material and methods.

Lines 300-302: This is a repetition of the material and methods

Lines 302-312: The inhibition area reported in Figure 1 are very small, inconspicuous and sometimes not present.

Table 2. is an image with low quality. Please insert a table not as a picture. All the concentration used must be reported. The test must be performed at least 3 time.

Author Response

The manuscript describes the probiotic characterization and the safety assessment of Lactic Acid bacteria isolated from raw milk of Japanese-Saanen Goat (Capra hircus). The paper is very interesting, but it must be improved. Is necessary to repeat all the tests at least 3 times to bee scientifically valid.

[AU] The authors thank you for reviewing our manuscript. Your valuable comments and suggestions are highly appreciated.

Simple Summary

Line 19: Staficolococci are not enteropathogenic bacteria. I suggest changing the terms in antibacterial activity.

[AU] All the term “enteropathogenic” have been changed to “pathogenic” throughout the text.

Line 20: “L. rhamnosus” must be written in italics

[AU] It is corrected accordingly.

Abstract

Line 30: see the comment reported for line 19.

[AU] It is corrected accordingly.

Keywords: “anti-enteropthogenic activity” see the comment reported for line 19

[AU] It is corrected accordingly.

Introduction

Lines 66-70: Not a clear sentence, please re-write it.

[AU] It is re-written as “Fermented foods are undoubtedly the best source for exploring beneficial LAB. Indeed, several LAB strains have been isolated to date from fermented foods, such as pickles [30,31], beverages [32,33], yogurts [25,34–36], cheeses [25,37–40], and sausages [41].”

Materials and Methods

Line 107: “with 2% glucose” insert data on the glucose used (Producer, Town, State), here and for all the media/substance and reagent used.

[AU] To indicate producers’ information for all the general reagents is not supposed to be suitable, a sentence “General chemicals used in this study were analytical grade.” is inserted in line 104.

Line 108. “in anaerobic conditions” insert data on the anaerobic conditioner.

[AU] The sentence “To generate anaerobic atmosphere, Anaeropack Kenki (Mitsubishi Gas Chemical Co., Inc., Tokyo, Japan) was used.” is added in lines 109-110.

Line 109, 111, etc… : See the comment reported in line 107 “phosphate-buffered saline (PBS)”,”cycloheximide”, bromocresol purple (BCP), ect… please check in all the paper. (Line 299,300, 304, etc.…)

[AU] Please see the response above.

Lines 114-115: no clear procedure, please explain better.

[AU] The sentence is rephrased as “To obtain pure colonies, single-colony isolation was performed for all isolates.”

Line 116: “..in 30% glycerol until use” insert the media used as 70%.

[AU] It is rephrased as “…in 30% glycerol containing MRS until use.”

Line 133: “Lacticaseibacillus rhamnosus” change in italics

[AU] It is corrected accordingly.

Lines 164-191: see the comment reported for line 19.

[AU] It is corrected accordingly.

Lines 166-170: The used strain are wild strains or certified. Please insert some other information.

[AU] The words “all the clinical isolates” are inserted in lines 172-173.

Line 175: “105 to 106 CFU/mL” please verify

[AU] The “5” and “6” should be in superscript, so it is corrected.

Lines 185-186: “The agar plates were then incubated at 4°C for 1 h…” please explain the reason of this passage.

[AU] This is the time for CFCS to diffuse into agar. The words “to diffuse CFCS into the agar” are added in line 189.

Lines 164-191: Is not reported the number of repetitions performed. In the Lines 193-277: for all the experiments performed are not reported the repetition (at least 3 must be performed). So, I think that they have performed only one time, therefor the results reported are not significant.

[AU] We have performed three independent experiments for all the experiments required to obtain quantitative data in this study, such as successive treatment of low pH and bile salts, antibiotic susceptibility test, antimicrobial activity test, and plasminogen binding/activation test, and hence subsection 2.13. is added. Sentences “Data was analysed statistically by Student’s t test. Significant difference was indicated by asterisk.” is added in the legend of Figure 1. Sentences “The concentrated CFCS of strain YM2-3 showed significantly higher anti-pathogenic activity than that of YM2-1, regardless of neutralisation treatment. The neutralisation of concentrated CFCSs significantly decreased and abolished antipathogenic activity against S. sonnei and L. monocytogenes, respectively.” are added in Lines 319-322 in subsection 3.2 accordingly.

Results

Table 1: is an image with low quality. Please insert a table not as a picture. The description of the procedure reported in the Material and methods is not equal to those reported in the Table. In the Table 1, no anaerobic condition is reported only for MRS; and for M17 is reported only 1 temperature 42°C in anaerobic and 30°C in aerobic. Please explain better or modify the material and methods.

[AU] Table 1 is replaced with the one in word format. We examined all the conditions described in the Materials and Methods, but results are indicated only the conditions that colonies were emerged. Therefore, a sentence “Results of the culture conditions that colonies emerged were indicated.” is added as the footnote of the Table 1.

Lines 300-302: This is a repetition of the material and methods

[AU] This sentence is deleted.

Lines 302-312: The inhibition area reported in Figure 1 are very small, inconspicuous and sometimes not present.

[AU] To support understanding the data, a new Table 2 is added. Therefore, sentences “Data was analysed statistically by Student’s t test. Significant difference was indicated by asterisk.” is added in the legend of Figure 1. Sentences “The CFCS of strain YM2-3 showed significantly higher anti-pathogenic activity than that of YM2-1, regardless of neutralisation treatment. The neutralisation of CFCSs significantly decreased and abolished antipathogenic activity against S. sonnei and L. monocytogenes, respectively.” are added in Lines 319-322 in subsection 3.2. A sentence “Representative images were shown.” is inserted in the legend of Figure 2.

Table 2. is an image with low quality. Please insert a table not as a picture. All the concentration used must be reported. The test must be performed at least 3 time.

[AU] Table 2 is replaced by Table 3 in word format. Since no variation among the triplicated data in the antibiotic susceptibility test was observed, we didn’t perform statistical analysis for this experiment. A sentence, “The same MIC values were obtained for three independent experiments.” is added in line 340.

Round 2

Reviewer 2 Report

The authors have revised the manuscript accordingly. It can be considered for publication. 

Reviewer 3 Report

The authors have imporved the paper with all the suggestion/request.